# Pain Management in Farm Animals: Focus on Cattle, Sheep and Pigs

**DOI:** 10.3390/ani11061483

**Published:** 2021-05-21

**Authors:** Paulo V. Steagall, Hedie Bustamante, Craig B. Johnson, Patricia V. Turner

**Affiliations:** 1Department of Clinical Sciences, Faculty of Veterinary Medicine, Université de Montréal, 3200 Rue Sicotte, Saint-Hyacinthe, QC J2S 2M2, Canada; 2Veterinary Clinical Sciences Institute, Faculty of Veterinary Sciences, Universidad Austral de Chile, Independencia 631, Valdivia 5110566, Chile; hbustamante@uach.cl; 3Animal Welfare Science and Bioethics Centre, School of Veterinary Science, Tāwharau Ora, Massey University, Palmerston North 4472, New Zealand; c.b.johnson@massey.ac.nz; 4Global Animal Welfare and Training, Charles River, Wilmington, MA 01887, USA; pvturner@uoguelph.ca; 5Department of Pathobiology, University of Guelph, Guelph, ON N1G 2W1, Canada

**Keywords:** analgesia, animal welfare, cattle, cost-benefit, ethics, pain, pig, sheep

## Abstract

**Simple Summary:**

Pain causes behavioral, autonomic and neuroendocrine changes and is a common cause of animal welfare compromise in farm animals. These recommendations focus on cattle, sheep, and pigs, and present the implications of unmanaged pain in terms of animal welfare and ethical perspectives, and its challenges and misconceptions. We provide an overview of pain management including assessment and treatment applied to the most common husbandry procedures, and recommendations to improve animal welfare in these species.

**Abstract:**

Pain causes behavioral, autonomic, and neuroendocrine changes and is a common cause of animal welfare compromise in farm animals. Current societal and ethical concerns demand better agricultural practices and improved welfare for food animals. These guidelines focus on cattle, sheep, and pigs, and present the implications of pain in terms of animal welfare and ethical perspectives, and its challenges and misconceptions. We provide an overview of pain management including assessment and treatment applied to the most common husbandry procedures, and recommendations to improve animal welfare in these species. A cost-benefit analysis of pain mitigation is discussed for food animals as well as the use of pain scoring systems for pain assessment in these species. Several recommendations are provided related to husbandry practices that could mitigate pain and improve farm animal welfare. This includes pain assessment as one of the indicators of animal welfare, the use of artificial intelligence for automated methods and research, and the need for better/appropriate legislation, regulations, and recommendations for pain relief during routine and husbandry procedures.

## 1. Introduction

Around the world, changing moral and ethical concerns have meant that society is demanding better agricultural practices and improved welfare for food animals. In this context, appropriate pain management is a fundamental pillar for improving the welfare of farm animals. Pain causes behavioral, autonomic, and neuroendocrine changes. It induces a negative affective state and is a common cause of animal welfare compromise. For example, chronic pain can reduce food consumption and average daily weight gain, increase heart rate and blood pressure, and decrease body temperature in cattle [1]. However, pain is still neglected, under-recognised, and under-treated in farm animals. This article summarises the implications of pain in farm animals (focusing on cattle, sheep, and pigs) in terms of animal welfare and ethical perspectives, and its challenges and misconceptions. We provide an overview of current pain management practices, including assessments and treatments for common procedures. Additionally, recommendations on the use of alternative methods for painful cosmetic procedures, improved education of veterinarians and farmers on pain management, as well as critical future research needs are discussed with the goal of providing guidance for stakeholders, the veterinary profession, and ultimately improving farm animal welfare.

## 2. The Implications of Pain: Animal Welfare and Ethical Perspectives

The International Association for the Study of Pain defines pain as “An unpleasant sensory and emotional experience associated with, or resembling that associated with, actual or potential tissue damage” [2]. Mammals generally have similar nociception across different species, and it is safe to assume that events that are painful in humans are experienced similarly in other mammals [3]. The basis for most animal welfare frameworks, such as the Five Freedoms [4] or the Five Domains [5], is that animals are sentient and can suffer, and are thus worthy of human respect and care [6], and moral consideration [7]. This duty of care concept also underlies most animal welfare legislation (see for example, the 2007 Lisbon Treaty [8]). However, which animals are protected and how much protection is afforded to them varies considerably between countries. 

Under recognition and treatment of pain are widespread in human and veterinary medicine despite decades of research in this area [9,10,11]. While pain can exist without suffering [12] and vice versa, failure to adequately manage pain is a significant cause of suffering in humans as well as in animals, and farm animals are no exception. Most veterinary association oaths recognise the professional duty of veterinarians to relieve pain and suffering. Yet, there is widespread veterinary acceptance of painful conditions and procedures in farm animals [13]. This issue is linked to societal norms for a given country or region; cost and convenience of treating pain in food animals; education and awareness of pain occurrence and mitigation; as well as alternatives to painful practices, availability of licensed products for use in farm animals, and individual (i.e., producer and veterinary) ethical frameworks for managing and caring for animals and promoting their welfare [2]. When viewing this issue globally, it seems to be an immense problem, and this has resulted in complacency and a general failure to hold veterinarians and animal producers accountable for better management and mitigation of pain in farm animals. The problem is shared amongst different stakeholders including the veterinary profession, farmers, associations, governments, and industry. Additionally, animal welfare may not be prioritised because of insufficient resources in low to middle income countries, and there may be a lack of reflection by veterinary professionals on the ethical and animal welfare implications of long entrenched agricultural animal practices, such as mulesing, nose ring placement, tail docking, etc. In other cases, there is a lack of robust evidence for the efficacy of approved therapeutics to manage pain for certain procedures, such as castration of piglets. However, even in countries in which licensed and efficacious pharmaceutical products are available, their use can be low [14] and is often not mandatory for the specific animal industry. 

Veterinary practice is somewhat modeled on principalism [15], an ethical framework widely used in human medicine. That is, veterinarians are generally guided by concepts of autonomy (of the client), beneficence, nonmaleficence (do no harm), and justice. However, principalism fails as an ethical framework for veterinarians (and human physicians) when there are conflicts between principles, for example, when conducting a procedure without analgesia in an animal (nonmaleficence) is requested by the client (autonomy) [16]. When placed in these conflicts, it can be difficult for the veterinarian to resolve their ethical duty and it may be easier to default to the path of least resistance (i.e., what the client desires). 

Decision-making in veterinary practice might be better guided by a care-based ethical framework [10] or by virtue ethics [17]. Because traditional normative ethical frameworks, such as consequentialism (e.g., utilitarianism) and deontology are rationally based, they generally do not take into account the emotional and interactive components of everyday veterinary practice. And yet, a care-based applied ethical approach allows the practitioner to defend some emotions, such as compassion, as being moral and important in ethical decision-making [18]. With a care-based approach, the veterinarian’s duty of care would be more closely oriented to the animal or patient, which is more vulnerable and more likely to suffer in the case of procedures, such as castration and dehorning, deemed necessary for raising animals for food consumption. Virtue ethics also recognises the primacy of emotions to moral perception and those subscribing to this ethical framework must try to act virtuously in their interactions with others [17]. Using a virtual ethics approach for pain management in farm animals, the needs of both the animal and the client would be recognised and the veterinarian would feel compassionate towards both parties. However, within this ethical framework, protecting the most helpless agent (the animal) would still be the most just course of action. With both frameworks, the veterinarian’s duty to protect and care for animals is clear and should drive their actions. 

While regulatory frameworks are helpful for advancing the care and management of pain in farm animals, regulations often lag behind societal mores. Veterinarians, in particular, need to step forward to accept increasing accountability for ensuring that alternatives to painful animal husbandry practices are used, whenever possible, and when there are no alternatives, that procedures are conducted competently, use the least invasive technique possible, and that appropriate, evidence-based pain mitigation strategies are used, when available. This requires an appropriate ethical framework for professional practice and is complimented by the development of increased training and education strategies for farm animal producers.

## 3. Neurophysiology of Pain, Stress, Fear and Anxiety

An affective state is a mental construct that can best be described as what it is like to feel a particular emotion or feeling [19]. The interaction between a particular affective state and its underlying physiological mechanisms is complex [20]. The physiological mechanisms underlying pain are referred to as nociceptive pathways [21]. While this delineation can be a useful aid to understanding the interaction between physical mechanisms and affective states, the two are co-dependant and so intimately associated as to make the distinction almost artificial.

Nociceptive pathways (i.e., transduction, transmission, and modulation until perception of noxious stimuli) form a complex interaction of ascending and descending neurological pathways supported by an array of chemical regulators such as neurotransmitters and hormones [22]. While the anatomical pathways are an important component of the system, the majority of analgesic strategies target its chemical regulation, and this is critical in terms of clinical management of pain. Minor species differences exist for nociceptive function, but major regulators [23] are present across species. The interaction between physiological activity (e.g., nociception) and affective state is well defined for pain.

However, other modifiers of affective state, such as anxiety and distress, can also alter the perception of physiological stimuli and reduce thresholds for pain perception [24]. As a consequence, perceived pain is more severe in animals that are subjected to negative affective states including prolonged pain perception due to painful conditions, such as lameness [25], or interventions such as castration [26]. Prolonged pain perception also promotes other negative affective states, such as anxiety and depression [27]. This association between increased pain perception and altered affective state can lead to a downward spiral of worsening pain perception and welfare compromise [28]. Box 1 presents a simple glossary for better understanding of the neurophysiology of pain. 

Box 1Glossary of terms used in the neurophysiology of pain.*Peripheral sensitisation*—reduced pain thresholds that occur following tissue damage that is caused by an in-crease in the concentration of chemical mediators in the damaged tissue. There are many mediators of peripheral sensitisation, and as a group they are called eicosanoids [21].*Central sensitisation*—reduced pain thresholds that occur following tissue damage that is caused by altered sensitivity of the nociceptive pathways because of repeated stimulation. The main sites of central sensitisation are the dorsal horn of the spinal cord and the higher centres of the brain, especially the cerebral cortex.*Hyperalgesia*—increased pain due to stimulation of damaged tissue by a stimulus that would normal-ly be perceived as painful. The stimulation would normally be painful, but it is felt as more painful than it would be if the tissues were not damaged. It is a clinical sign of peripheral and central sensitisation.*Allodynia*—increased pain due to stimulation of damaged tissue by a stimulus that would not normally be perceived as painful. The stimulation would not normally be painful, but it is felt as painful because the tissues are damaged. It is a clinical sign of peripheral and central sensitisation.*Persistent postoperative pain*—pain perceived following surgery that lasts longer than it usually would. It is a common consequence of poor or absent pain relief at the time of surgery.

## 4. Challenges in Pain Management and Misconceptions

The provision of pain management to farm animals raises a number of legal and practical challenges. In the legal sense, there is a tension between the competing needs of the animals and society as a whole. Legislative frameworks for the administration of analgesic drugs to these animals need to take, for example, the potential for abuse if potent controlled analgesic drugs, such as narcotics, are not properly stored and accounted for, and the potential for consumer harm if drug residues remain in products derived from the animals that are intended for human consumption. In a practical sense, the cost of the drugs themselves, together with the cost of complying with their associated legal requirements, can reduce the profitability of the farms and result in animals that would benefit from pain relief remaining untreated (Section 5). Alternatives to husbandry practices to improve welfare should take this financial burden into consideration. 

This tendency towards undertreatment of pain in farm animals is compounded by the estimated value of a given animal. High value animals (for example, semen providers for artificial insemination) are more likely to receive treatment for painful conditions because of their high economic worth. They also tend to have longer lives, making withdrawal periods less problematic. Animals with lower value are more likely to live shorter lives and to contribute directly to food products, making the relative cost of pain relief more significant and the legislative complexities of using analgesic drugs more problematic.

Where large numbers of animals are kept in intensive husbandry systems, routine veterinary care can be focussed on the system as a whole rather than on the individual animals in that system. In these situations, particular care should ensure that there is sufficient oversight of the individual animals to be able to identify and treat those that may be suffering from painful conditions. In situations where direct veterinary care is not practicable, those people caring for the animals should be given training on pain recognition, and systems should be put into place to enable pain management to individuals. The advent of artificial intelligence and machine learning may help identifying painful individuals in intensive systems with a large number of animals.

As mentioned, several misconceptions exist in the use of analgesics in food animals. It is often believed that they feel less pain than other mammals [29]; that financial constraints do not allow the use of analgesics in these species [30,31,32]; and that young calves, lambs or piglets may not feel pain as adult animals do [33]. Undeniably, the lack of education on the subject in the veterinary curriculum, the lack of recognition of species-specific pain-induced behaviors [34], the fear of analgesic-induced adverse effects, and milk and meat production withdrawal times after drug administration contribute to the lack of pain management in food animals [35]. For example, the prevalence of analgesic administration is higher in horses than bovids [31], and pigs receive less analgesics than horses and cattle [31,34]. One study demonstrated that less than 0.001% of pigs received analgesia after castration [31]. On the other hand, food animal pain management has evolved in recent years with the advent of pain scoring systems including facial expressions; the understanding of pain behaviors; and the study of analgesic techniques on blood, clinical and production parameters, and affective states (e.g., pain, fear, negative cognitive bias) (see Section 5). New methods, approaches, and technology for farming may also help with pain mitigation. For instance, interaction with humans and handling during the first 2 weeks of life decreased pain sensitivity in lambs and could be integrated to sheep farming [36]. Similar findings have been observed in pigs [37].

## 5. Pain Assessment and Recognition in Farm Animals 

Pain recognition and detection are the only means for developing appropriate therapy and understanding whether treatment is effective. Pain recognition in farm animals (i.e., cattle, sheep, and pigs) can be challenging due to their stoic and prey behaviors. Often these animals avoid demonstrating vulnerability as species. Additionally, pain recognition is difficult because veterinarians are not always appropriately trained on farm animal pain management, thus pain-related behaviors are not always recognised [38]. In recent years, the field has evolved with the advent of pain assessment scoring systems/tools including composite scales and facial expression scales for acute pain recognition with reported validation (Table 1). Facial expressions have been used for pain assessment in sheep with pododermatitis and mastitis [39], and the Sheep Grimace Scale [40]. A recent validated sheep composite scale for the assessment of abdominal pain also has been published [41]. Facial expressions of pain have also been identified in cattle [42], with the advent of the Cow Pain Scale and the UNESP-Botucatu unidimensional bovine composite pain scale [35] for assessing postoperative pain, as well as a Piglet Grimace Scale [43,44] and a facial expression scale in sows [39]. A recent composite scale has also been published for acute pain assessment in pigs [45] (Table 1). 

Where there is a lack of validated tools for pain assessment in farm animals, pain may be evaluated with behavioral scoring for that condition [51,53] (e.g., numerical descriptive scoring systems for lameness in cattle or sheep). This can be supplemented with clinical and blood biomarkers as well as production parameters. Changes in behavior for cattle in pain include a lack of interaction with the environment, increased vocalisation, reduced locomotion, changes in body posture and hind limb positions (e.g., after castration), reduced activity, lowered head position, and increased attention to wound [35]. Changes in facial expressions can be observed and used for pain assessment (Table 1). 

Painful sheep may present with reduced sociability, decreased food consumption, tremors, abnormal vocalisation, lameness or altered gait and locomotion, and potentially, changes in facial expressions (Table 1) [36,38,54,55,56]. Healthy pigs are usually curious animals that demonstrate complex behavior [57]. Growing pigs that are painful may demonstrate decreased interactions with their pen mates and the environment. They may present cognitive dysfunction under stress with early weaning and isolation, leading to increases in the prevalence and duration of abnormal behaviors [58,59]. These changes in cognitive and normal behaviors include less activity, lying, sleeping, eating, drinking, and sniffing, and reduced agonistic interactions with their mates. The latter behaviors can also be observed in painful pigs, showing that pain and distress are often correlated [60]. Increased aggressive behavior, social isolation, and changes in posture or stereotype behaviors have also been reported in painful pigs, and these changes commonly disappear with the administration of analgesics [60,61,62,63,64]. Indeed, one study showed that the administration of local anesthesia for castration has positive effects on weight gain at weaning and 1 week after the procedure [65].

Pain assessment and recognition may be more challenging with a large number of animals kept on industrial farms. A regular program for observing and monitoring animals each day for overall condition should be in place with the training of personnel to detect animals demonstrating signs of pain or ill health. Other alternatives for improving farm animal health and indicators of welfare should be considered. For example, accelerometers and pedometers have been used to monitor mobility [66]. These inexpensive and easy-to-use devices may be employed for detecting behaviors that could be related to negative affective states including pain. Similar tools have been used in sheep for lameness detection [67]. Accelerometers and pedometers provide objective timing of recumbency, standing, and activity [68]. Using these devices, it has been determined that adult cattle spend more time standing after orchiectomy [69] and calves spend more time lying after castration or dehorning [70]. The use of artificial intelligence may also provide interesting means of automated pain assessment using machine learning in the future. For example, a pain analyzer system prototype and facial expression scales have been reported in sheep [71].

## 6. Cost-Benefit Analysis of Pain Mitigation in Farm Animals

Appropriate management of pain in farm animals can improve long-term productivity, reduce animal suffering, and blunt stress responses during a painful procedure [72], while representing best husbandry practices. An understanding of how economic factors affect how farm animals are treated will help policy makers to improve animal welfare, enabling accountability for losses associated to painful husbandry procedures [73]. Several economic aspects of farm animal pain management have been studied, including trade, incentive programs, cost-benefit, and consumer preferences [74]. The magnitude of the cost differs according to the country, cultural beliefs, type of animal production, farm type, and availability of technology [75]. The use of the term ‘harm’ instead of ‘cost’ is also encouraged so pain management is not to be thought of as a purely economic concept [76]. Veterinarians should take these perspectives into consideration when determining whether to use analgesics for routine invasive husbandry procedures. Consumers are often willing to pay for products originating from farms with higher animal welfare standards [77]. Thus, pain management may be driven by ethical considerations and consumer demand rather than by purely economic reasons [78]. 

While not necessary to justify the ethical imperative for using analgesics or anesthetics for painful procedures, some studies in farm animals have identified potential gains in animal production when analgesics are administered. The use of systemic analgesia increased daily gain weight by 0.97 kg and the concurrent use of systemic analgesia and cornual nerve block by 1.0 kg [74] in calves undergoing disbudding. Similarly, the use of meloxicam in calves undergoing disbudding increased daily weight gain for up to 15 days compared with calves that did not receive analgesia [79]. Local anesthetic block of the spermatic cord during surgical castration of dairy calves increased daily weight gain by 30% up to day 35 post-castration [80]. Likewise, a 96% increase in daily weight gain was observed after administration of ketoprofen and a local anesthetic block before castration [81]. The administration of local anesthesia in combination with a sedative (i.e., xylazine) and a nonsteroidal anti-inflammatory drug (NSAID) (e.g., ketoprofen, meloxicam) improved average daily weight gain after dehorning compared to control calves (without the administration of NSAID) [82]. Play behaviors were more evident in calves undergoing hot-iron disbudding and receiving an NSAID than controls [83]. Increased feed intake after disbudding has been reported in calves that received an NSAID [84]. Thus, in a number of cases, a clear cost-benefit has been demonstrated for the use of analgesics for painful procedures in food animals.

## 7. Treatment of Pain in Farm Animals for Routine Procedures

### 7.1. Castration

Castration is a painful procedure that is routinely performed on young male animals of multiple species, including pig, beef and dairy cattle, and sheep. The general purpose is to decrease aggression, improve ease of handling and housing, and to prevent unwanted breeding. Castration is also performed specifically to reduce boar taint in pigs and improve palatability of meat from cattle and sheep, all of which are caused by accumulations of androsterone and skatole in the muscle of intact, sexually mature males [85]. Castration can be performed by several means, some of which are species-specific. Commonly employed techniques include surgical castration with removal of testicles via an open incision (all species); transcutaneous crushing of the spermatic vessels and cord with a tool (emasculator or Burdizzo) with eventual sloughing of the scrotum and testicles (cattle, sheep); and banding (rubber rings), in which a tight rubber ring is placed around the scrotum and testicles, resulting in sloughing of ischemic and necrotic tissue in 3–6 weeks (cattle, sheep). 

All methods of castration at any age in any species are painful [86,87,88]. There is minimal evidence of analgesia efficacy for surgical castration in pigs with any of the currently approved NSAIDs or local anesthetics, such as lidocaine [52,89,90,91], and some evidence of partial efficacy for analgesia following castration by band or surgical means in beef cattle [92,93]. In general, there is a lack of information about analgesia efficacy for surgical castration of sheep and few licensed analgesic drugs available for use [94] in this species. In one study, an analgesic benefit was observed after the administration of morphine and flunixin meglumine [45]. One promising analgesia technique that has been recently described is the use of lidocaine-loaded castration bands in 3–4-week-old beef calves [92]. This method provides sustained transcutaneous release of local anesthetic within the scrotal tissues while the band is in place. Surveys of animal producers suggest that when analgesic use for castration is not mandated by legislation, use of analgesics for castration is low [95]. Because of societal concerns [96,97] regarding pain in food animals undergoing castration, this is an active field of research including alternatives to surgical castration. In some countries like Norway, pig castration is no longer allowed without the administration of local anesthetics [98]. The procedure must be performed by a veterinarian in that country. 

In some countries, such as the UK and New Zealand, male pigs are marketed at an immature size, significantly reducing the need for castration [99]. Other techniques to avoid surgical castration include immunocastration [100,101] of males with a gonadotropin releasing hormone vaccine and gene editing [102,103] of commercial pigs to reduce or remove production of substances responsible for boar taint. Ram lambs should be left intact whenever possible; short-scrotum procedure should be carried out if surgery is required [104]. Similarly, lambs for human consumption must not be tail docked. The above examples are potential avenues that may eliminate the need for surgical castration of hundreds of millions of pigs on an annual basis around the world.

### 7.2. Mulesing

Mulesing is a painful husbandry procedure performed in lambs and involves surgically eliminating wrinkled skin from the perineal region to avoid cutaneous myiasis. Significant pain behaviors develop between 2–5 h after the procedure, characterised by hunched posture, still walking, and reduced lying and grazing behavior [105]. Pain from mulesing can last from weeks to months [106]. 

Pre-emptive use of a buccal formulation of meloxicam was reported to result in decreased pain-induced behaviors immediately after the procedure [107]. The use of a topical agent containing lidocaine, bupivacaine, epinephrine, and cetrimide has been shown to be effective at providing pain relief for 12–24 h following mulesing [108,109]. Similarly, the administration of injectable meloxicam and topical agents (lidocaine, bupivacaine, cetrimide and epinephrine) resulted in a reduction of plasma cortisol [110] and pain-induced behaviors during the 6 h following mulesing [111]. 

The replacement of mulesing by methods of genetic selection involving breeding of animals with fewer wrinkles in the perineal area has been a subject of research [104]. Non-surgical alternatives have been developed including the use of specifically designed occlusive plastic clips, the subcutaneous injection of specific chemical agents, and the development of integrated pest management practices as well as long-term insecticide agents and anti-fly vaccines [104,112]. 

### 7.3. Branding

Hot iron or freeze branding are used extensively to identify cattle and other animals managed in extensive productive systems. Approximately 57% of Western Canada farmers perform branding, with only 4% of cases reporting some form of pain mitigation [113]. Hot iron branding involves cauterizing the skin, while freeze branding uses liquid nitrogen, causing the destruction of melanocytes in the skin [104]. Both methods induce a marked pain-induced escape-avoidance reaction with a higher stress response induced by hot iron branding [114,115]. Moreover, hot iron branding induces long-term hyperalgesia lasting up to 2 months with prolonged healing time [116]. Although initially painful, freeze branding seems to cause less discomfort and pain-specific behavior, and represents a refinement to hot iron branding [117]. Animals would greatly benefit from replacing branding with newer identification technologies, including the use of body tags, leg sensors, neck chains/collars, and other technologies such as injectable transponders and rumen boluses. When alternatives are not available or possible, the use of analgesics must be considered for branding. The administration of meloxicam has shown to reduce pain behaviors during branding associated to castration. Therefore, the use of an NSAID is strongly recommended for the procedure [118]. 

### 7.4. Ear Notching/Tagging

Ear tagging and ear notching are husbandry procedures routinely performed in calves, lambs, and piglets. Tagging involves ear piercing to allow the placement of an identification tag [119]. Notching is the removal of part of the pinna following a defined pattern [117]. The ear is a sensitive organ, thus ear tagging and notching are painful procedures [120]. Inflammation, tears, and the risk of infection may increase pain associated with the procedure. Limited scientific information exists regarding how to reduce pain caused by these procedures. The use of a vapocoolant spray reduced pain-induced behavioral responses after ear tagging and notching in unweaned calves [119]. Pre-emptive administration of meloxicam reduced plasma cortisol after ear tagging in piglets, suggesting that systemic analgesics may be beneficial for ear procedures [121]. The use of radiofrequency identification devices represents a valid alternative for the replacement of ear tagging and notching [122]. These include subcutaneous, intraperitoneal or intraruminal sensors, and external devices such as tags and collars [123]. 

### 7.5. Tail Docking 

Tail docking is a painful procedure that is routinely performed on young animals of both sexes, including pigs and sheep, although for different reasons. Tail docking is routinely performed in piglets raised under intensive conditions to prevent tail biting, which can be a significant issue and result in moderate to marked wounding, pain, and spinal abscesses. The procedure induces acute behavioral and physiologic responses indicative of pain and stress responses including vocalization, increased sitting, and increased plasma cortisol levels [89,124,125,126]. The causes for biting are likely complex and have been reviewed elsewhere, but it may occur because of boredom or redirected exploratory behavior [99,127]. In piglets, tail docking is conducted in neonatal piglets with clippers or cautery. Both are painful procedures, although the use of hot cautery may be slightly less painful than clippers. The use of topical anesthetics with or without NSAIDs provides minimal pain relief [128] for tail docking in piglets. Ultimately, it is preferable to find better housing methods that eliminate the requirement for tail docking, as is done in some countries, such as Sweden [129].

Tail docking is routinely conducted in young lambs in warmer and more humid regions of the world to minimise the likelihood of flystrike (myiasis), a serious condition that can occur when faeces and other material accumulates under the tail, causing a moist dermatitis that can subsequently become embedded with maggots. In some UK surveys, flystrike may be seen on-farm with up to 80% during warm, humid months, with an average of 1.4% ewes and 2.8% of lambs affected [130,131,132]. Tail docking is commonly conducted either surgically or by banding, similar to castration procedures. Tails that are docked too short or improperly can be associated with rectal prolapse [133] or neuroma formation [134], respectively. The use of subcutaneous injections of topical anesthetic [e.g., lidocaine/bupivacaine mixtures] is required in some countries, such as the UK, after seven days of age, and is reported to be associated with a reduction in pain responses in lambs [135,136]. There are few alternatives to tail docking of sheep in areas where blowflies are endemic. Shearing ewes prior to warm, humid conditions may reduce the incidence [132], and some research has examined sheep with genetically shorter tails, which would not require docking, since tail length is highly heritable. This is an area in which further research is needed. 

Tail docking is performed in some countries for dairy cattle because it is thought to improve udder cleanliness. However, recent studies have demonstrated that tail docking and tail switch trimming does not improve udder cleanliness or reduce the incidence of mastitis in dairy cows and it can have severe adverse welfare outcomes for cows, which are no longer able to remove flies and other insects [137,138] with their tails. Veterinarians are strongly encouraged to re-educate dairy farmers to eliminate this practice altogether.

### 7.6. Dehorning/Disbudding

Disbudding consists of removing or destructing the horn bud before it is attached to the underlying tissue. The procedure is performed in very young calves and goat kids. Dehorning consists of removing the horn after its attachment to the underlying tissue. The procedure is performed to reduce risk of injuries to producers, handlers, and herd mates. Calves are still dehorned with little to no pain management in many countries [139,140]. Specific training of farmers to perform the procedure with analgesics is rarely provided, and veterinarians do not always administer local anesthesia for the procedure. For example, over 40% of veterinarians in the United States did not administer analgesics for dehorning in a recent study [141]. 

As observed during dehorning, hot-iron disbudding produces severe pain for hours as evidenced by severe burns and large open wounds, changes in behavior (e.g., vocalization, kicking and falling), decreases in mechanical nociceptive thresholds, and increases in serum cortisol levels [142,143]. Burns may produce inflammation and long-term peripheral sensitization [144]. Pain-induced behaviors in calves after dehorning include head shaking, ear flicking, head rub against surfaces and objects, frequent changes in position, increased time lying [145], and vocalization up to 72 h after the procedure [82,146]. The use of local anesthetics and NSAIDs before dehorning will blunt these responses [142,147]. For example, the use of a cornual nerve block will decrease head and leg movements during dehorning [142]. The cornual block should also be supplemented with subcutaneous infiltration of a local anesthetic around the horn basis (ring block). The use of a cornual nerve and “ring block” with an NSAID preoperatively for disbudding and dehorning in calves [148] has been demonstrated to relieve pain. The nerve block will also prevent short- and long-term increases in serum cortisol levels and heart rate, respectively [142]. Dehorning adult cattle should include the administration of a multimodal protocol (xylazine, local anesthesia, and an NSAID) [147,149]. 

Breeding high genetic merit polled cattle (cattle that do not develop horns) may be considered as an alternative to dehorning/disbudding, while carefully avoiding relatedness and inbreeding in the population [150]. Additionally, small farmers who did not perform dehorning reported no difficulty in handling horned cattle [151], and alternatives to the procedures must always be considered.

### 7.7. Nose Ringing

Nose ringing refers to insertion of a metal ring through the nasal septum in cattle to facilitate control of animals, especially bulls. Similarly, it is used in sows under extensive production systems to reduce and prevent digging and rooting [152]. In sows, one ring should be preferably positioned in the nasal septum instead of multiple rings through the cartilage at the top of the snout [104]. If this procedure is to be implemented, it must be performed at the earliest possible age. Nose rings must be placed with the animal securely restrained and sedated, preferably with the use of local anesthesia and proper systemic analgesic treatment. An infraorbital nerve block may be used for placement of a nose ring in calves. The administration of the block in piglets may be challenging and stressful for the animal, and the administration of NSAIDs is recommended instead. The infraorbital block emerges through the infraorbital foramen, rostral to the facial tuberosity, where it is covered by the nasolabialis muscle. A variable amount of local anesthetic (10–20 mL) is deposited in the emergence of the nerve at both sides of the face [153]. Alternatives to this procedure include maintaining lower densities of animals [154] and feeding diets with a higher percentage of fiber [155].

## 8. Recommendations and Future Research

Many painful practices are established into the production of various food animals around the world. Increasingly, there is recognition of the sentience of these animals as well as enhanced public interest and scrutiny into these procedures and overall concern with farm animal welfare. Cost, convenience, and the lack of a specific approved product and approved drug withdrawal time cannot justify withholding analgesics for painful procedures. Several national veterinary associations and statutory bodies have provided guidance to their members regarding selection and use of an appropriate analgesic when no registered product exists [156,157]. This helps to ensure that veterinarians can prioritise animal welfare when making decisions for the animals in their care.

The following represent general recommendations for improving pain recognition and analgesia management in food animals:There is a need for better pain mitigation strategies in farm animals undergoing husbandry procedures and a critical need to improve animal welfare. Some management practices require further discussion, including research on alternatives to eliminate painful animal husbandry practices. For example, new identification technologies, including the use of painless body tags, body sensors, neck chains/collars, and microchip transponders should be considered for animal identification. This would avoid branding, and ear tagging and notching.When alternatives are not available, husbandry procedures should be conducted competently, using the least invasive technique possible and evidence-based analgesic techniques and pain mitigation strategies. This requires an appropriate ethical framework for professional practice.Veterinarians should be better educated on farm animal pain management. Course curricula should include pain assessment and recognition, the use of pain scoring systems and pain mitigation strategies, and implications of husbandry practices on animal welfare.Increased training and education of farm animal producers should be provided. Veterinarians should be more engaged with client education. Some farmers are not always aware of analgesics used for pain management or the potential consequences of husbandry practices for their animals.Better means of pain management are needed with the study of novel analgesic techniques and therapeutics, as well as their impact on production costs, gains, and other parameters.Pain assessment should be included as one of the indicators of animal welfare. Research studies should incorporate the use of validated pain scoring systems in farm animals.The use of artificial intelligence for automatic pain assessment and recognition may be a promising and objective alternative for pain assessment of farm animals.There is currently a gap in appropriate legislation, regulations, and recommendations for pain mitigation during routine and invasive husbandry procedures in farm animals. These discussions should involve veterinarians, producers, associations, societal and governmental bodies, and the industry, among many stakeholders.

## 9. Conclusions

Around the world, many painful husbandry practices are conducted routinely on farm animals. Attention to this source of animal suffering is an imperative for veterinarians, animal producers, legislators, and others.

## Figures and Tables

**Table 1 animals-11-01483-t001:** Summary of pain assessment composite tools and facial scales.

Species	Instrument	Focus	Description	References
Bovine	UNESP-Botucatu unidimensional pain scale for acute postoperative pain assessment in cattle	General behaviors	Developed in beef cattle (2–3 years-old) undergoing castration. Includes five items (locomotion, interactive behavior, activity, appetite, miscellaneous behaviors) scored from 0 to 2 for a total score of 10. Analgesic threshold: >4/10 The UNESP-Botucatu unidimensional pain scale for acute postoperative pain assessment in cattle has also been validated in Italian.	**[35]**, [46]
	Cow Pain Scale	General behaviors and facial expressions	Developed in adult dairy cattle (unclear age) with naturally-occurring medical or surgical painful conditions. Includes six items. Attention towards the surroundings and facial expression are scored from 0 to 1. Head position, ears position, response to approach, and back position are scored from 0 to 2 for a total score of 10. An analgesic threshold (3/10) is subjectively suggested. However, it has not been defined based on statistical calculations.	[42]
Sheep	The UNESP-Botucatu sheep acute composite pain scale (USAPS)	General behaviors	Developed in adult sheep (3.5 ± 1.8 years-old) undergoing elective laparoscopy. Includes six items (interaction, locomotion, head position, posture, activity, appetite) scored from 0 to 2 for a total score of 12. Analgesic threshold: ≥4/12.	[41]
	Sheep Grimace Scale	Facial expressions	Developed in adult laboratory sheep (unclear age) undergoing tibial osteotomy. Includes three items. Orbital tightening and ear & head position are scored from 0 to 2. Flehmen is scored from 0 to 3 for a total of 7.	[40]
	Sheep Pain Facial Expression Scale	Facial expressions	Developed in sheep (>1 year-old) with footrot or mastitis. Includes five items (abnormal ear position, orbital tightening, abnormal nostril and philtrum shape, cheek tightening, abnormal lip and jaw profile) scored from 0 to 2 for a total of 10. No analgesic threshold available.	[47]
	Lamb Grimace Scale	Facial expressions	Developed in lamb (5–6 weeks-old) undergoing tail docking. Includes five items (ear position, orbital tightening, nose changes, cheek flattening, mouth change) scored from 0 to 2. No analgesic threshold available.	[48]
Swine	UNESP-Botucatu pig composite acute pain scale (UPAPS)	General behaviors	Developed in growing pigs (38 ± 3 days-old) undergoing castration. Includes six items (attention to affected area, locomotion, activity, appetite, interactive behaviour and miscellaneous) scored from 0 to 3 for a total of 18. Analgesic threshold: ≥6/18.	[45]
	Sow Grimace Scale	Facial expressions	Developed in sows (unclear age; gilts and multiparous sows) undergoing farrowing. Includes five items (tension above eyes, snout angle, neck tension, temporal tension and ear position, cheek tension) scored from 0 to 2 for a total score of 10. No analgesic threshold available.	**[44]**, [49,50]
	Piglet Grimace Scale (PGS)—a	Facial expressions	Developed in piglets (5 days-old) undergoing castration. Includes three items. Ears and cheek bulging/nose bulge are scored from 0 to 2. Orbital tightening is scored from 0 to 1 for a total of 5. No analgesic threshold available. Obs 1: The PGS was validated in growing pigs (73 ± 11 days-old) undergoing castration and laparotomy due to unilateral cryptorchidism [51]. Obs 2: In two studies assessing the responsiveness of the PGS to buprenorphine, meloxicam and ketoprofen in piglets undergoing castration [52,53]; the item ‘ears’ were scored from 0 to 3 for a total score of 6.	[50]
	Piglet Grimace Scale—b	Facial expressions	Developed in piglets (3 days-old) undergoing tail docking and castration. Includes seven items (temporal tension, forehead, eyes, tension above the eyes, cheek, snout plate, snout angle, lip, jaw, and nostril) scored from 0 to 2. Each item is evaluated independently and there is no final score. No analgesic threshold is available.	[43]

References in **bold** refer to the study reporting the development and initial validation of the scale. Additional references refer to subsequent studies.

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
