# Peer review of "Pain Management in Farm Animals: Focus on Cattle, Sheep and Pigs"

_animals, 2021, doi:10.3390/ani11061483_

Round 1

Reviewer 1 Report

Reviewer comments on Manuscript number: animals-1221411

The present manuscript shows a review about pain recognition, assessment and treatment in cattle, sheep and pigs, considering the importance of producer-farmer and Doctors of Veterinary Medicine attitudes to pain management in above species.

The manuscript is well written, and the information showed is interesting and relevant, as its main aim is to provide current information about the sometimes neglected pain mitigation in farm animals.

Broad comments

The manuscript is well written, and the information showed is valuable and relevant, just few comments and suggestions before it could be published.

In simple summary and abstract you referred to small ruminants in general, but you focused on sheep. You may choose to add information about goats in the entire document, or stay only with sheep as expressed in the title.

To this reviewer, adding information about pain management in goats could be of great value to this paper.

Thank you

Author Response

The present manuscript shows a review about pain recognition, assessment and treatment in cattle, sheep and pigs, considering the importance of producer-farmer and Doctors of Veterinary Medicine attitudes to pain management in above species. The manuscript is well written, and the information showed is interesting and relevant, as its main aim is to provide current information about the sometimes neglected pain mitigation in farm animals.

Answer: Thank you very much for your comments

Broad comments

The manuscript is well written, and the information showed is valuable and relevant, just few comments and suggestions before it could be published.

In simple summary and abstract you referred to small ruminants in general, but you focused on sheep. You may choose to add information about goats in the entire document, or stay only with sheep as expressed in the title.

To this reviewer, adding information about pain management in goats could be of great value to this paper.

Answer: Thank you for your suggestion. The review is already long so the consensus of our group was to focus on pain management in sheep. We have now replaced ‘small ruminants’ with ‘sheep’ in the abstract.

Reviewer 2 Report

Very interesting topic from the ethical point of view. Even the functional balance of the foot in cattle must be treated and above all the alternative to lidocaine, a drug used for 90% in the surgery of farm animals.

 In this regard, I suggest some interesting articles:

Costa Giovanna, Musicò Marcello , Spadola Filippo ,Cortigiani Sergio ,.Leonardi Fabio , Cucinotta Giuseppe, Interlandi Claudia 2018. Effects of tramadol slow injection vs fast bolus in the therapeutic balance of the foot in bovine. Large animal review 24: 19-21

Interlandi, C., Nastasi, B., Morici, M., Calabrò, P.,Costa, G.L.2017 Effects of the combination romifidine/tramadol drug administration on several physiological and behavioral variables in calves Large Animal Review 23:51-54 2

Spadola, F.,Costa, G.L., Morici, M., Interlandi C., Nastasi, B., Musicò, M.2017 Autologous prosthesis for the surgery of two simultaneous hernias in a calf Large Animal Review 23:195-197 1

Author Response

Very interesting topic from the ethical point of view. Even the functional balance of the foot in cattle must be treated and above all the alternative to lidocaine, a drug used for 90% in the surgery of farm animals.

 In this regard, I suggest some interesting articles:

Costa Giovanna, Musicò Marcello , Spadola Filippo ,Cortigiani Sergio ,.Leonardi Fabio , Cucinotta Giuseppe, Interlandi Claudia 2018. Effects of tramadol slow injection vs fast bolus in the therapeutic balance of the foot in bovine. Large animal review 24: 19-21

Interlandi, C., Nastasi, B., Morici, M., Calabrò, P.,Costa, G.L.2017 Effects of the combination romifidine/tramadol drug administration on several physiological and behavioral variables in calves Large Animal Review 23:51-54 2

Spadola, F.,Costa, G.L., Morici, M., Interlandi C., Nastasi, B., Musicò, M.2017 Autologous prosthesis for the surgery of two simultaneous hernias in a calf Large Animal Review 23:195-197

Answer: Thank you for your suggestion. Pain assessment for lameness in cattle is indeed discussed in the article. However, the review article is already long (6500 words) and we did not include specific treatments for alleviating pain in cattle with limb pathologies. The literature provides extensive information on the subject including specific review articles and an issue of Vet Clin North Amer dedicated to ‘Lameness in Cattle’.

https://pubmed.ncbi.nlm.nih.gov/22558967/

https://pubmed.ncbi.nlm.nih.gov/23438403/

https://pubmed.ncbi.nlm.nih.gov/24560824/

https://pubmed.ncbi.nlm.nih.gov/28392188/

https://www.sciencedirect.com/science/article/pii/S0749072017300178

https://pubmed.ncbi.nlm.nih.gov/30592750/

Reviewer 3 Report

This article talks about the pain management of farm animals, including the implications of pain, neurophysiology of pain, challenges in pain management and misconceptions, pain assessment and recognition, cost:benefit analysis of pain mitigation, treatment of pain for routine procedures, recommendations and future research. There was an abundance of content in this article. The article provides a lot of valuable information about pain management in farm animals.

But this article is too long (23 pages). How many readers have the patience to read the full text? I suggest that pain management should be introduced separately by animal species, so that interested readers can choose and read. Because the readers who pay attention to these three farm animals are different.

Title: delete the period

Line 26, “small ruminants” change to “sheep”

Line 336, “on young, male animals of”, change to “on young male animals of”

Line 337, “pigs” change to “pig”

References: there are some references from a URL, but I can not open them. Can you quote journal articles as much as possible?

Author Response

This article talks about the pain management of farm animals, including the implications of pain, neurophysiology of pain, challenges in pain management and misconceptions, pain assessment and recognition, cost:benefit analysis of pain mitigation, treatment of pain for routine procedures, recommendations and future research. There was an abundance of content in this article. The article provides a lot of valuable information about pain management in farm animals.

But this article is too long (23 pages). How many readers have the patience to read the full text? I suggest that pain management should be introduced separately by animal species, so that interested readers can choose and read. Because the readers who pay attention to these three farm animals are different.

Answer: Thank you for your comments. The review article is indeed long because the topic is extensive even when goats and other species have not been included, or when treatment of lameness is not discussed, for example. Such review articles can be used for consultation, training and as an overview of the issue as commissioned by the World Veterinary Association’s Animal Welfare Group. The article has been separated in sections so the reader can easily consult the manuscript in full or parts of it. We respectfully disagree that reading a long review article is about patience in this case. Additionally, the authors opted to present treatment of pain based on routine procedures instead of a species-specific approach, since the same procedure can be performed in different species. On the other hand, the reader can easily consult Table 1 for pain assessment by animal species as suggested by the reviewer. In terms of animal welfare, readers may want to learn about pain assessment for farm animals in general.

Title: delete the period

Answer: Deleted.

Line 26, “small ruminants” change to “sheep”

Answer: Changed.

Line 336, “on young, male animals of”, change to “on young male animals of”

Answer: Changed

Line 337, “pigs” change to “pig”

Answer: Changed

References: there are some references from a URL, but I can not open them. Can you quote journal articles as much as possible?

Answer: We tried to provide relevant references as much as possible in the review. The URLs are working when tested again. Some of the URLs have been removed and others have been mostly used for official links to European laws and news, USDA documents, or code of practices on issues related to animal welfare. Thank you.